# The Impact of the Chinese Thinking Style of Relations on Mental Health: The Mediating Role of Coping Styles

**DOI:** 10.3390/bs13060442

**Published:** 2023-05-24

**Authors:** Wanhe Meng, Minxuan Feng, Huihui Yu, Yubo Hou

**Affiliations:** Beijing Key Laboratory of Behavior and Mental Health, School of Psychological and Cognitive Sciences, Peking University, Beijing 100871, China; 1401110589@pku.edu.cn (W.M.); minxuan.feng@stu.pku.edu.cn (M.F.); huihuiyu@stu.pku.edu.cn (H.Y.)

**Keywords:** Chinese people, relational thinking, connectedness, mental health, coping style

## Abstract

Chinese people tend to view and analyze problems according to relations and holism, which can cause them to adopt positive coping strategies when facing difficulties, thus improving their mental health. This study verifies the relationship among relations as a dimension of the Chinese thinking style, coping strategies, and mental health through three research studies. Study 1 preliminarily examines a significant, positive correlation between Chinese relational thinking and mental health through questionnaire surveys. Study 2 primes Chinese relational thinking and explores its relationship with coping strategies. The results show that relational thinking could enhance individuals’ active coping, seeking of emotional support and venting, problem avoidance, and attentional diversion coping strategies while reducing denial and disengagement coping styles. Study 3 further demonstrates through questionnaires across time points that Chinese relational thinking could improve individuals’ mental health by enhancing their active coping and minimizing denial and disengagement. The results of the three studies are of great significance in terms of improving mental health from the perspective of Chinese relational thinking and coping strategies.

## 1. Introduction

For years, we have discussed the occurrence of distinctive cognitive styles due to cultural differences. Extensive research has shown that Easterners perceive and organize information differently than Westerners in general [1,2]. Easterners, primarily referring to the Chinese, approach problems by considering the situation or context, emphasizing continuity and relationships. Conversely, Westerners prefer to derive the attributes and characteristics of an object or phenomenon from the overall context and observe its nature [3,4]. Therefore, it is reasonable to assume that Westerners are more inclined to use logical methods, such as classification, deduction, and reasoning, to analyze and solve problems. By contrast, Easterners tend to consider the various effects of an event and strive for neutral and *Zhong Yong* (midway) solutions to avoid extremes [5,6].

The investigation conducted by Norenzayan and colleagues [7] carried out on the causal attribution of conflicts showed differences between Westerners and Easterners in how each group perceives the cause of an event. Easterners tend to believe that a specific event is caused by the interaction between an individual’s personality traits and the social context, while Westerners attribute the cause solely to personal disposition. This difference implies that East Asians prefer to use external attribution or to consider both internal and external factors, whereas Westerners tend to use internal attribution, believing that a person’s disposition determines what happens [8]. In another study conducted by Morris and Peng [9], similar results were obtained regarding cultural divergence in media reports concerning reports of the same criminal event. The analysis suggested that Chinese newspapers were apt to emphasize the influence of environmental factors, such as living conditions and strained interpersonal relationships, on a murderer’s behavior. Conversely, American newspapers tend to highlight the personality traits of murderers, such as being cold-blooded and having violent tendencies [1,9].

### 1.1. Chinese Thinking Styles

For the last two decades, cultural psychologies have focused on the features of the Chinese thinking style, as well as its underlying mechanism, as functioning distinctively compared to Westerners. Nisbett and colleagues [4] pinpointed a systematic difference in thinking styles between the Greeks and the Chinese. The unique epistemology influenced by metaphysics, which consists of belief in nature and individuals’ understanding to perceive knowledge, triggers people in Chinese culture to comprehend and cognize the world holistically, which covers scientific, mathematical, and philosophic fields [4] (see similar findings in [10,11]). Such thoughts in holism manifest as thinking persons and objects in forms of continuity, dialectics, and even as an experience base [12]. According to Hou et al. [3], the Chinese thinking style contains three differentiated dimensions: relations, change, and contradiction. Relations are defined as the close interconnection between things in the world; change means that the world is constantly developing and unstoppably changing; and contradiction refers to the status that things exist in opposition, and both sides coexist simultaneously, which infers a dialectical cognition (see similar findings in [13]). Chinese thinking styles emphasize the concept of interaction and conflict; hence, Chinese people are inclined to understand objects and situations by considering their contextual information, and they are willing to take in ideas from friends and families. Zhen-Dong and other colleagues [14] further highlighted that people who are educated under the Chinese educational system are more probably affected by Confucious values, which primarily entail harmonious and interdependent fulfillment. Such a value required a meta-cognition mode of thinking in holism and interconnection, navigating Chinese people to cognize more holistically and relational-interdependently.

Given the unique attributes of Chinese thinking styles, cultural psychologists have paid attention to the impact of thinking styles on psychological well-being. For instance, Rodgers et al. [15] studied the associations of dialectical thinking with self-evaluation, self-esteem, and mental health. Their study showed that contradictions in Chinese dialectical thinking could negatively affect an individual’s mental health by lowering levels of self-evaluation and self-esteem. Similarly, another study by Hou [16] found that the three dimensions of Chinese thinking (relations, change, contradiction), along with other cognitive traits, such as attribution style and responsibility attribution, could influence people’s psychological well-being and social adaptation. Interestingly, these experts proposed that the mechanisms of different dimensions in Chinese thinking styles are not entirely the same. For example, Luo [17] illustrated that the dimension of relation in dialectical thinking has a positive effect on participants’ mental health, while contradictory thinking negatively impacts mental health (see similar findings in [15,16,18].

The purpose of this study is to explore the associations between Chinese thinking styles and mental health. In light of the explanation of different dimensions and features of Chinese thinking, we concluded that the relational thinking style optimizes personal capabilities by considering problems regarding internal and external factors. Therefore, we believe that relational thinking is beneficial to mental health, especially when people experience setbacks and failures due to uncontrollable circumstances. Based on this belief, this study proposes the following.

**Hypothesis** **1.**
*Chinese relational thinking is significantly, positively correlated with mental health.*


### 1.2. The Mediating Role of Coping Styles

Coping and coping styles refer to individuals’ strategies and behaviors to manage and deal with stress, adversity, or challenging situations [19,20,21]. Coping involves cognitive and behavioral efforts and is often classified into different categories with distinct features. According to the traditional trait theory, coping styles denote that individuals possess a unique set of stable, enduring traits that are relatively consistent across situations and over time [22,23]. Based on this view, some researchers have developed dispositional coping questionnaires or coping style scales. A representative example is the Multidimensional Coping Inventory (COPE), developed by Carver et al. [24]. This questionnaire divides common coping strategies into two broad categories: problem-focused coping and emotion-focused coping. Each category can be further divided into various coping styles, including positive coping, planning, cognitive appraisal, denial, avoidance, and psychological disengagement.

By contrast, another famous hypothesis, “situation theory”, emphasizes the role of context in shaping individual understanding when managing stress and issues [25,26]. It rests on the idea that people adapt to different coping strategies for different situations. Based on the above two different perspectives, Folkman et al. [27] proposed a dynamic mechanical model of person–environment interactions: namely, the cognitive appraisal theory for stress and coping. This theory emphasizes the mediating role of coping between stress and coping outcomes, suggesting that coping styles are influenced by both personality and situational factors. As a result, the outcomes of coping styles (positive or negative) further affect the individual and environmental conditions, promoting continuous changes and the development of coping styles through dynamic interactions [27,28]. Therefore, different coping styles and strategies can affect an individual’s health status and psychological adaptation.

For quite some time, Western researchers have recognized the significance of the relationship between coping strategies and various variables, such as physical health, psychological symptoms, and well-being [27,29,30]. For instance, McCrae and Costa [31] conducted a study exploring the operational mechanisms of coping styles and personality traits, such as neuroticism, extraversion, and openness. Their study examined adaptive (such as active coping, seeking support, emotional expression, and cognitive reappraisal) and maladaptive coping styles (such as passivity, wishful thinking, and self-blame). McCrae and Costa [31] found that adaptive coping was effective at solving problems and relieving emotions and was associated with the openness personality trait. In contrast, maladaptive coping was ineffective and associated with neuroticism personality traits. This outcome suggests that personality traits can predict and influence an individual’s coping strategies and psychological health, indicating that coping strategies might have cross-situational consistency. Kobasa [32] also reported that proactive coping strategies were generally associated with extroversive personality traits and innovative cognitive styles. However, passive coping strategies occur due to cognitive biases and specific thinking patterns that impede problem-solving and long-term mental health [33].

According to prior research on coping and coping styles, individuals with different dispositions and cognitive styles can conduct different coping strategies regarding their levels of bearing stressors [29,34]. A few Chinese studies have illustrated that the Chinese relational thinking style plays a vital role in predictions of more active coping with less denial and fantasy [35,36]. Additionally, relational thinking is also closely associated with psychological adaptation and is able to forecast levels of psychological health [3,17]. Based on the theoretical analysis and the results of previous studies, we believe that there is a hypothetical relationship among associative thinking, coping styles, and psychological health as follows.

**Hypothesis** **2.**
*Coping styles play a mediating role between Chinese individuals’ relational thinking and mental health. Specifically:*


**Hypothesis** **2a.**
*Active coping strategies can mediate the relationship between relational thinking and mental health, meaning that relational thinking promotes positive coping strategies, which, in turn, improve psychological health;*


**Hypothesis** **2b.**
*Denial and disengagement coping can mediate the relationship between relational thinking and mental health, implying that relational thinking reduces denial and disengagement coping and enhances psychological health;*


**Hypothesis** **2c.**
*Seeking emotional support and venting coping can mediate the relationship between relational thinking and mental health, implying that relational thinking promotes seeking emotional support and venting, which, in turn, improves psychological health;*


**Hypothesis** **2d.**
*Problem avoidance and attentional diversion coping can mediate the relationship between relational thinking and mental health, meaning that relational thinking promotes problem avoidance and attentional diversion coping, consequently enhancing psychological health.*


### 1.3. The Present Research

This study validated the hypotheses proposed through three different research studies. First, we used a questionnaire to explore the relationship between relational thinking and mental health in Study 1. Next, in Study 2, we explored the relationship between relational thinking and coping strategies by conducting a priming experiment on Chinese people’s relational thinking. Finally, we examined the relationship among Chinese relational thinking, coping strategies, and mental health through a longitudinal study in Study 3.

## 2. Study 1

This study aimed to examine the relationship between Chinese relational thinking styles and mental health through self-reported questionnaires.

### 2.1. Methods

#### 2.1.1. Participants

The experiment was conducted in a laboratory in the form of in-person questionnaires comprising 209 valid collections. All participants were from communities that were interested in the research field. Each survey was designed with two attention check questions with mandatory questions to be answered, such as “Please choose ‘strongly disagree’ to pass the attention check test”. Participants who accomplished all the questions and passed two attention check questions would be paid 4 RMB (Chinese Yuan). The sample consisted of 111 men and 98 women, with an average age of 29.51 years old (SD = 5.09). Seven questionnaires were excluded due to either answering all items with the same option or not passing the attention check questions, which were included in the Chinese relational thinking style questionnaire and required participants to select “strongly disagree” for that particular item.

#### 2.1.2. Measure of Chinese Thinking Style on Relations

To test individual degrees of Chinese relational thinking, we adopted the Structure of the Chinese Thinking Style, which was developed by Hou [16] and has been prevalently used when examining Chinese holistic thinking. The scale is composed of three dimensions—Relation, Contradiction, and Change—representing three aspects of the Chinese thinking style. Our study utilized the subscale of Relation, which comprised 4 items measured on a 7-point Likert scale in terms of each question. The subscale measurement was composed of questions such as “Do you believe things that perform in complete non-relations are subtly interconnected and relational-independent” and “Do you think when a person changes oneself, the person will change surrounding people”. The 7-point rating scale ranged from 1 (strongly disagree) to 7 (strongly agree), indicating a gradual extent of recognition. In this study, the internal consistency coefficient of the Relational Thinking Style subscale was 0.69 (M = 5.43, SD = 0.73).

#### 2.1.3. Measure of Mental Health

The study measured mental health using the General Health Questionnaire (GHQ) 20, which was developed by Goldberg [37] and adapted by Li and Mei [38] in China. The GHQ-20 consisted of 20 items measuring three dimensions: self-affirmation (e.g., “Are you able to concentrate on what you are doing?”), depression (e.g., “Do you feel you have failed in life?”), and anxiety (e.g., “Do you constantly feel under strain?”). Participants were asked to indicate whether they had experienced each phenomenon over the previous few weeks, and responses were measured on a “yes” or “no” scale. This study calculated the mental health score as the self-affirmation subscale score minus the depression and anxiety subscale scores (M = 3.38, SD = 1.63).

#### 2.1.4. Measure of Control Variables

The control variables in this study were age and gender, which were demographic variables.

### 2.2. Results

#### 2.2.1. Common Method Bias Test

Since all the data in the study were reported by the participants simultaneously, a common method bias test was conducted first. The present study used Harman’s single-factor test to test for common method bias [39]. All items were included in one factor, and the rotated factor analysis showed that the cumulative variance explained by 11 factors with eigenvalues greater than 1 was 61.00%. The largest eigenvalue factor explained only 13.17% of the variance, which was less than the boundary of 40%, indicating that the common method bias in the data did not seriously affect the results of this study.

#### 2.2.2. Correlation Analysis

We conducted a correlation analysis between the Chinese relational thinking style and mental health. After controlling for age and gender, the Chinese relational thinking style positively and significantly correlated with mental health (r = 0.14, *p* < 0.05). Additionally, the regression analysis showed that, after controlling for age and gender, the Chinese relational thinking style was positively associated with mental health: B = 0.71, t = 2.03, *p* < 0.05. These results indicate that the relationship between Chinese thinking style, especially the Relation dimension, and mental health was positively significant, which supported Hypothesis 1. The results remained unchanged after controlling for the control variables.

## 3. Study 2

The present study was targeted to explore the causality of Chinese relational thinking and coping styles. We established a priming experiment to stimulate participants’ concepts of interconnection in Chinese thinking and followed it with self-reported questionnaires to examine individual levels of coping.

### 3.1. Methods

#### 3.1.1. Participants

Data collection for this study was conducted through the Chinese online platform Credamo, which is a frequently used Chinese software that is designed for psychological data collection and duration by pulling questionnaires. All participants were anonymous. The sample consisted of 300 participants, including 140 men and 160 women, with a mean age of 27.57 years old (SD = 2.88). Five invalid questionnaires were removed due to the detection of identical responses for all items as well as for participants who failed the attention check questions, which were included in the coping style questionnaire and required participants to select “strongly disagree”.

#### 3.1.2. Measures and Procedures

Our study used a single-factor complete randomized design. This study was designed in the parts of the priming experiment, coping style scale, and demographic information collection. Participants were first asked to read the informed consent form and were then randomly assigned to the interconnection group (*N* = 150) or the analysis group (*N* = 150). The reference to the priming experiment was derived from previous research on holistic thinking styles [40]. Participants in the interconnection group were asked to focus on the overall background of the picture and write a story based on the content of the picture, while participants in the analysis group were asked to identify and depict as many animals as possible were nested in the picture. Afterward, the experimenter presented another memory picture to the participants and asked them to memorize it for 90 s. Then, the memorizing materials were presented to the participants, who were asked to recall the items at 10 locations corresponding to their respective positions in the memory picture as a manipulation check. Subsequently, participants were asked about their willingness to adopt coping strategies, as well as their demographic information. After completing a series of experimental procedures and passing the attention check, participants received a cash reward of 3 RMB.

The researchers adapted the coping style scale that Shi et al. [21] developed by changing the original prompt—“Everyone encounters difficulties or problems in daily life. These difficulties may be related to work or study, family, interpersonal relationships, or one’s health. The test includes descriptions of coping strategies that people may adopt when encountering different difficulties. Please circle how often you use these strategies according to your actual situation”—to “How willing are you to adopt these coping strategies when solving problems?”

### 3.2. Results

#### Manipulation Check

The independent samples from the *t*-test revealed that individuals in the interconnection group (M = 3.32, SD = 0.45) had better recall performance than those in the analysis group (M = 3.14, SD = 0.45; t = 3.20, *p* < 0.01, Cohen’s d = 0.33), indicating the successful priming of relational thinking.

As shown in Figure 1, the independent samples *t*-test further demonstrated that, in terms of willingness to adopt positive coping strategies, individuals in the interconnection group (M = 3.32, SD = 0.45) scored significantly higher than those in the analysis group (M = 3.14, SD = 0.47; t = 3.38, *p* < 0.01, Cohen’s d = 0.39). Regarding the willingness to seek emotional support and venting, participants in the interconnection group (M = 4.06, SD = 0.47) also gave a significantly higher score than those in the analysis group (M = 3.47, SD = 0.47; t = 7.81, *p* < 0.001, Cohen’s d = 0.09). Similarly, in light of problem avoidance and attentional diversion strategies, the participants in the interconnection group (M = 3.64, SD = 0.46) scored significantly higher than those in the analysis group (M = 3.26, SD = 0.48; t = 5.37, *p* < 0.001, Cohen’s d = −0.62). By contrast, the willingness to deny and disengage showed the opposite tendency. Individuals in the interconnection group (M = 3.18, SD = 0.44) scored significantly lower than those in the analysis group (M = 3.47, SD = 0.50; t = −5.48, *p* < 0.01, Cohen’s d = −0.63). These results indicate that people with the relational thinking style tend to adopt more positive coping strategies and avoid maladaptive ones.

## 4. Study 3

The present study aimed to further explore whether the coping style can function solidly as an intermediating role of the correlation between Chinese relational thinking and psychological well-being. According to Study 2, Chinese relational can promote individual capabilities of active coping, seeking emotional support and venting, problem avoidance, and attentional diversion coping while soothing denial and disengagement coping strategies. Hence, we collected self-reported questionnaires at two different time points to observe whether the mediating role of the above four coping styles existed.

### 4.1. Methods

#### 4.1.1. Participants and Procedure

This study distributed two questionnaires through the online platform Credamo at two different time points, with valid collections of 300 questionnaires and 256 questionnaires separately. The first questionnaire consisted of the subscale of Relation from the Structure of the Chinese Thinking Style [16] and Coping Style Scale [21], and the second questionnaire used the General Health Questionnaire (GHQ) 20 [37,38]. Both questionnaires were designed with attention check questions with mandatory questions such as “Please choose ‘strongly disagree’ to pass the attention check test”. Nine invalid collections were excluded due to not passing the attention check questions. The sample included 112 men and 114 women, with a mean age of 27.97 (SD = 7.55).

#### 4.1.2. Measure of Chinese Thinking Style of Relations

The measurement of Chinese relational thinking used in this study was the same as in Study 1. The internal consistency coefficient of the scale in this study was 0.60.

#### 4.1.3. Measure of Coping Style Scale

We employed the same measurement of coping styles as Study 2 in this paper. The internal consistency coefficient of the active coping subscale was 0.60; the denial and psychological disengagement subscale was 0.78, the emotional expression and ventilation subscale was 0.68, and the avoidance and distraction subscale was 0.73.

#### 4.1.4. Measure of Mental Health

The measurement of mental health utilized in this study was the same as in Study 1.

#### 4.1.5. Measure of Control Variables

The control variables in this study were age and gender among the demographic variables.

### 4.2. Results

#### 4.2.1. Common Method Bias Testing

Although the data in this study were collected across different time points, all data were self-reported by the participants. Therefore, this study conducted a common method bias test. The study used the human single-factor test method [39] to test for a common method bias, and all items were placed in the same factor. The results showed that the cumulative variance explained by the factor analysis with 28 eigenvalues greater than 1 was 71.60%, with the largest eigenvalue explaining 19.34% of the variance, which was less than the boundary value of 40%. This outcome suggested that a common method bias in this study did not have a significant impact on the results.

#### 4.2.2. Correlation Analysis

Table 1 shows the means, standard deviations, and correlation coefficients of the main variables in this study.

#### 4.2.3. Mediation Analysis

The overall model was tested using Model 4 in the process plugin in SPSS software, version 26.0. As shown in Figure 2 and Table 2, the results illustrated that, after controlling for gender and age, Chinese relational thinking was able to influence mental health through active coping (direct effect = 0.08, SE = 0.03, 95% CI = [0.04, 0.15]) alongside denial and psychological disengagement coping strategies (direct effect = −0.04, SE = 0.02, 95% CI = [−0.08, −0.001]). The findings supported Hypotheses 2a and 2b. However, the coping strategies of seeking emotional support and venting (direct effect = 0.004, SE = 0.02, 95% CI = [−0.03, 0.04]) with problem avoidance and attentional diversion (direct effect = −0.01, SE = 0.02, 95% CI = [−0.05, 0.02]), respectively, did not have significant effects on mental health. Hypotheses 2c and 2d were, therefore, not supported. The results remained the same after removing the control variables.

## 5. General Discussion

This research examined the correlation between Chinese relational thinking and mental health in three studies, along with the mediating role of coping methods. According to the results of Study 1, consistent with previous research, Chinese relational thinking could be significantly associated with mental health (see similar findings in [4,41,42]. Previous studies have shown that individuals with solid relational thinking abilities tend to view the context and background as a whole, emphasizing the relationship between objects and contexts. This approach allows them to interpret and predict possible events based on the observed interconnections [4]. According to Luo’s research [17], individuals with solid relational thinking skills tend to contextualize particular problems or troubles without being trapped by limited comprehension regarding issues [43]. Additionally, Lu [44] found that individuals with higher levels of relational thinking tended to exhibit lower levels of depression and anxiety and were more inclined to express their demands reasonably to avoid unnecessary conflicts [42]. Thus, individuals with high levels of relational thinking tended to exhibit better psychological health since they could positively evaluate situations and consider problems from multiple angles.

Referring to Studies 2 and 3, our findings discovered the mediating role of coping styles in the association between relational thinking and mental health. This was consistent with prior studies, under the mediation of more active coping, as well as less denial and disengagement; Chinese relational thinking showed a positive correlation with mental health [21,45]. According to Shi et al.’s [21] explanation of the development of the scale, active coping emphasizes proactive problem-solving, which aims at effectively addressing difficulties. People with manageable active coping abilities are more apt to seek assistance from others or utilize a comprehensive thought process. In comparison, denial and disengagement coping exacerbated maladaptive coping approaches, involving indulging in fantasies, avoiding reality, and relinquishing proactive coping. Another meta-analysis by Liao [45] indicated that positive coping had a modest, negative correlation with psychological symptoms, while negative coping had a moderate, positive correlation with mental performance. In other words, adopting active coping and avoidance of maladaptive coping are beneficial for building a good sense of health concerns.

Nevertheless, this study also revealed that the coping strategies of seeking emotional support and problem avoidance were not found to have any significant impacts on mediating the relationship between relational thinking and mental health. This finding was also confirmed by Shi et al. [21] in the discussion. Shi supposed that seeking emotional support and problem avoidance should not be roughly determined as either absolutely positive or negative coping styles, but rather, should be understood regarding their effects on mental well-being when nested in specific contexts or situations. Similarly, in Liao’s study [45], emotional venting and attention-shifting coping were described as mixed coping strategies, implying a complex defining issue and a slight, positive correlation with psychological symptoms.

## 6. Implication and Future Studies

The research had profound practical implications with regard to enhancing Chinese people’s psychological well-being in terms of political and societal efforts. First, our study provided a brand-new vision of understanding and solving psychological health concerns when referring to Eastern psychological theories and government targeting when constructing a public welfare system. For the past three years, Chinese society has suffered unimaginably not only from the pandemic itself but also from the secondary economic ache, especially on adult employment. Considering the excessively intense competition in the Chinese job market lately, many young people report that they have encountered intensive stress derived from workplaces and everyday lives, such as family pressures and working involution, which refers to unnecessary competition for the purpose of promotion and higher salaries [46,47]. Such harsh burdens cause young adults to feel difficulties when managing working stress and frustration, resulting in puzzling behaviors and negative emotional regulation [48].

To prevent young people from disparate mental health deterioration, the Chinese government has advocated the construction of a social psychological service system to provide psychological help for communities and institutions when facing personal or social stress. This conceived construction requires the timely monitoring and resolution of individuals who require related psychological assistance and improvements in social governance [49,50]. To continuously promote the construction of the social-psychological service system, comprehensive coverage of the age range of psychological services must be achieved [49]. Hence, the present study fills a gap in the concerns of the mental development of young adults and provides thoughts and suggestions regarding the fostering of young people’s abilities to think in interconnection to the construction of a social psychological service system. Secondly, this study confirmed the impact of the Chinese relational thinking style on mental health. Combined with prior studies that examined the negative guidance of contradictory thinking styles on psychological health concerns [3], it was apparent that advocating relational thinking and avoiding contradictory thinking could be feasible when individuals undergo hardships and depression. Therefore, future studies could focus on exploring practical and concrete approaches to such advocation and avoidance thinking while also demonstrating specific ways or checklists on which people can rely in terms of improving mental well-being. However, it should be noted that individuals’ three dimensions of thinking styles develop in a distinctive pattern across ages [3,16,41]. Previous studies have revealed that the cognitive styles demonstrated in Chinese adult participants were distinguishable compared to adolescents [41]. Hou’s study [41] reported that adolescents did not perform differently regarding the levels of the three dimensions of thinking styles. In contrast, adults display a parallel development in both relational and contradictory thinking with changes in thinking style, performing in a different trend and showing a self-identity tendency similar to Westerners’ cognitive mode [41]. In addition, Hou and others [3,16] found that Chinese cognitive mode growth demonstrated peaks in specific age periods. Individuals’ thinking of connectedness gradually increased until the age of approximately 30 and then showed a slow decline. Similarly, the peak of contradictory thinking was between the ages of 18 and 22, with a downward trend afterward. Therefore, when improving mental health from the perspective of thinking styles, attention should be given to the difference in cognitive development according to age categorizations.

Third, our research discovered the significance of coping styles when influencing the impact of thinking style on psychological well-being. Therefore, in future efforts to improve the mental health of Chinese people, a starting point could be intervening in their coping strategies. Previous research has shown that factors such as age [51,52], gender [53,54], personality [31], cognitive factors [35], family factors [41], occupational factors [55], and social factors [55,56] all have corresponding effects on coping strategies. Therefore, influential factors could be selected from the above factors for intervention. At the same time, the government and society could actively guide individuals’ coping strategies through news, newspapers, or social media channels. For example, news or WeChat public accounts could provide information on the benefits of positive coping strategies, the disadvantages of negative coping strategies, and how to form positive coping strategies.

## 7. Limitations

While our study has made significant contributions to the existing literature on Chinese relational thinking and psychological well-being, it is important to acknowledge certain limitations. Firstly, due to constraints within our research field, our study was unable to gather samples from various geographical locations. To provide a more comprehensive understanding, future research should aim to include participants from different regions, thereby broadening the demographic representation of the sample. Secondly, although our study aimed to recruit participants spanning from different age groups, the collected sample primarily consisted of individuals within a narrow age range of approximately 26–30 years old. Consequently, the generalizability of our findings may be limited. Future studies should strive to diversify age groups and include participants from various academic levels, such as children, adolescents, and older people. Thirdly, our current research focused specifically on the influence of the Chinese thinking mode on mental health. However, it is crucial for future studies to explore whether cultural differences in thinking have varying effects on coping styles and psychological well-being. When acknowledging these limitations, future research endeavors could provide more comprehensive insights into the topic at hand, allowing for a deeper understanding of the interplay between cultural thinking patterns, coping mechanisms, and psychological well-being.

## 8. Conclusions

Our present study examined the relationship between Chinese relational thinking and mental health, as well as the intermediating role of coping styles in this correlation. This study illustrated that the relational thinking style displayed a significantly positive association in relation to mental health. What is more, the result suggested that active coping strategies aggressively promoted the positive correlation between interconnection thinking and psychological well-being, whereas denial and disengagement maladaptive coping triggered a negative effect in terms of such a correlation. Our study called on a robust national concern and perception in light of the importance of cultivating the relational thinking of Chinese citizens, particularly young Chinese adults.

## Figures and Tables

**Figure 1 behavsci-13-00442-f001:**
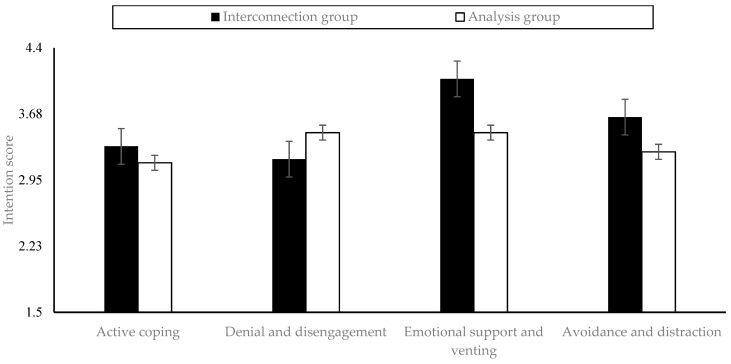
Differences in four coping strategies between the interconnection and analysis groups.

**Figure 2 behavsci-13-00442-f002:**
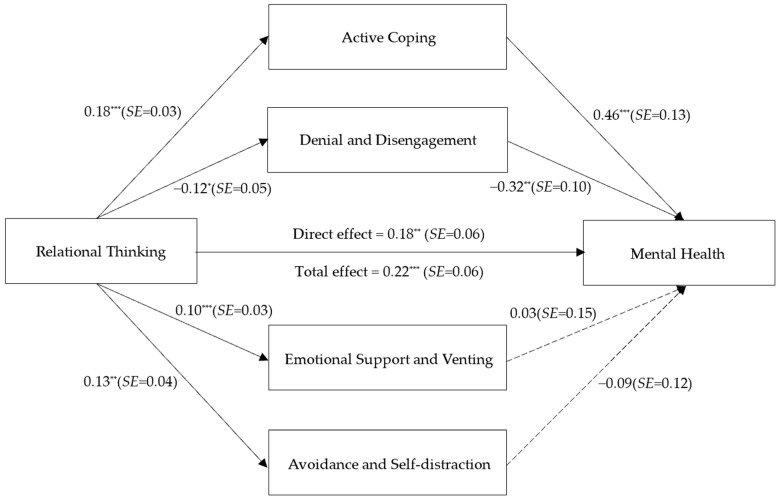
Mediation effect analysis. * *p* < 0.05; ** *p* < 0.01; *** *p* < 0.001.

**Table 1 behavsci-13-00442-t001:** Correlations between the main variables in Study 3.

	M	SD	1	2	3	4	5	6	7
1 Gender	-	-	1.00						
2 Age	27.97	7.55	−0.11	1.00					
3 Relational thinking	5.20	0.70	−0.02	0.13 *	1.00				
4 Active coping	3.22	0.37	−0.13 *	0.22 **	0.37 **	1.00			
5 Denial and Disengagement	1.99	0.58	0.17 **	−0.16 **	−0.12 *	−0.19 **	1.00		
6 Emotional support and venting	2.54	0.33	0.04	−0.16 *	0.20 **	0.21 **	0.52 **	1.00	
7 Avoidance and distraction	2.20	0.50	0.12	−0.18 **	0.16 **	−0.16 *	0.75 **	0.52 **	1.00
8 Mental health	−1.81	0.71	−0.10	0.32 **	0.25 **	0.41 **	−0.35 **	0.10	0.02

* *p* < 0.05; ** *p* < 0.01.

**Table 2 behavsci-13-00442-t002:** Mediation effects.

	Effect	SE	95%LLCI	95%ULCI
Connectedness → Active Coping → Mental Health	0.08	0.03	0.04	0.15
Connectedness → Denial and Disengagement → Mental Health	−0.04	0.02	−0.08	−0.001
Connectedness → Emotional Expression and Venting → Mental Health	0.004	0.02	−0.03	0.04
Connectedness → Avoidance and Distraction → Mental Health	−0.01	0.02	−0.05	0.02
Total Indirect Effect	0.04	0.04	−0.03	0.12

## Data Availability

The data are available from the corresponding author on reasonable request.

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
