# Peer review of "The Impact of the Chinese Thinking Style of Relations on Mental Health: The Mediating Role of Coping Styles"

_behavsci, 2023, doi:10.3390/bs13060442_

Round 1

Reviewer 1 Report

on the attach file 

Author Response

Please see the attachment. I have included my cover letter together with the point-by-point response. Many thanks!

Reviewer 2 Report

Overall, interesting research on the impact of the Chinese thinking style of relations on mental health. Some edits I recommend addressing. 1: It is unclear why there are three separate studies in this article. Why 3 separate studies and how do they relate? Especially since these were three separate samples at three difference time points. Please explain why. 2. Why the comparison between Western and Eastern Thinking styles at the beginning if there is no mention of impact on coping and mental health? The information just  hangs out there and does not tie into the rest of the paper. It seems the paper could just start with Chinese thinking styles - what case are you making to the reader about why it is important to know differences in thinking. Also it seems you are telling the reader about the difference in context as is appears in media reports but there are many western theories and applications of mental health in context and in relationships (social work, family therapy models, etc, systems theory, etc). 3). What is the attention check and why is this important? What is it ruling out and who does it rule of your participants? How many participants were excluded from your studies? Where was your sample from? 4).  In your discussion you present your findings as one study (unless I am understanding this wrong) but you conducted three separate studies which is important to address and breakdown. 5) Few if any limitations were mentioned. There was little age range represented, little demographic information presented, what this a representative sample? etc. 6). There are times when a number for a citation was given but the author was not listed and other times when it was... for example line 36 [7] has no author name but line 43 Morris and Peng [9] does. Check throughout. 

Author Response

Please see the attachment. I have included the cover letter together with the response to the comment. Many thanks!
